# Face cells encode object parts more than facial configuration of illusory faces

**Saloni Sharma** ✉, **Kasper Vinken**, **Akshay V. Jagadeesh & Margaret S. Livingstone**

Humans perceive illusory faces in everyday objects with a face-like configuration, an illusion known as face pareidolia. Face-selective regions in humans and monkeys, believed to underlie face perception, have been shown to respond to face pareidolia images. Here, we investigated whether pareidolia selectivity in macaque inferotemporal cortex is explained by the face-like configuration that drives the human perception of illusory faces. We found that face cells responded selectively to pareidolia images. This selectivity did not correlate with human faceness ratings and did not require the face-like configuration. Instead, it was driven primarily by the "eye" parts of the illusory face, which are simply object parts when viewed in isolation. In contrast, human perceptual pareidolia relied primarily on the global configuration and could not be explained by "eye" parts. Our results indicate that face-cells encode local, generic features of illusory faces, in misalignment with human visual perception, which requires holistic information.

One major goal of vision neuroscience is to understand the link between neural activity in the brain and visual perception. A central approach in this regard has been to interpret what individual neurons respond to. The groundwork for this notion of interpretable neurons was laid by Hubel and Wiesel[1,2], who proposed that cells at early stages of the visual system encode basic features like edges and orientations. At higher levels of visual processing, the discovery of neurons with selective responses for complex visual stimuli such as faces, hands, or objects in the inferotemporal (IT) cortex[3–7] led to the suggestion that these neurons directly underlie the perception of high-level categories[4,8]. However, a challenge with the approach of interpreting what a neuron responds to is that it is inherently subject to the biases of human perception itself. That is, when a neuron responds to faces, it may not necessarily reflect how we perceive faces.

For instance, decades of psychophysical experiments indicate that humans and other primates perceive faces holistically[9–11]: Humans recognize face identities more easily when the parts are embedded in the context of a whole face (whole-parts effect[12]); and when faces are presented upright rather than inverted (face inversion effect[13]). The dependence of face perception on holistic context was also demonstrated effectively with the Thatcher illusion[14], where humans and macaques recognize changes in the orientation of facial features (such

as eyes) only in the context of an upright, but not inverted, face ([15], but see also[16]). At the neural level, some of the hallmark properties of configural or holistic face processing have been observed in the responses of face-cells in IT cortex, the putative neural substrate of face processing[7,17–23]. For instance, scrambling faces into many smaller parts significantly reduces face-cell responses, which suggests a holistic dependence on facial configuration[3,5]. Further, face-cell responses are reduced to inverted faces (face inversion effect[24,25]) or when the eyes in an upright face are inverted (thatcher illusion[26]). On the other hand, it is less clear whether face cells demonstrate the whole-parts effect. One study showed that the gain of tuning in face cells for a particular feature was twice as high when presented in combination with other features in a face-like arrangement[25,27]. However, other studies showed that face cells respond to isolated face parts without requiring the context of a whole face[28,29]. Thus, while holistic processing plays a role in face perception, its role in the activity of face cells is less clear.

An interesting perceptual phenomenon thought to primarily rely on holistic processing is when illusory faces are perceived in inanimate objects, known as face pareidolia, such as "Jesus in toast" or "Man in the moon". The perception of illusory faces has been attributed to the configural processing of individual object parts that are spuriously

Department of Neurobiology, Harvard Medical School, Boston, MA, USA. ✉e-mail: Saloni_Sharma@hms.harvard.edu

arranged in the T-shaped global configuration (two eyes and a mouth), typically associated with faces[30–32]. The importance of the face-like configuration is highlighted by the fact that normal facial features are absent in these images and are replaced by notably heterogeneous features clearly incongruent with real faces[31]. Various human studies have quantitively demonstrated that such illusory faces are perceived as highly face-like[30,31,33,34]. In monkeys, eye-tracking experiments show that free-viewing monkeys look preferentially at pareidolia images, and that their fixation patterns for pareidolia images are correlated with the T-shaped viewing patterns for real faces[23,31]. Moreover, such images have been shown to engage face selective regions in both humans and monkeys[8,23,32,33,35–38]. These responses, thought to underlie rapid face detection, are suggested to be driven primarily by a template-matching process, where the template represents a rudimentary form of low or mid-level visual features that approximate a face-like configuration[8,33,35,39–41]. Taken together, it seems parsimonious to interpret neural selectivity for pareidolia images, defined by a template face-like configuration, as evidence for holistic neural processing that mirrors perception. However, for such images that are inherently subject to a perceptual bias (i.e., seeing faces where none exist), it may be particularly challenging to interpret what a face-neuron may be responding to.

Are responses to illusory face stimuli informative about the processing of actual faces? Recent evidence suggests that non-face stimuli engage the same mechanisms in macaque face cells as actual faces do[42]. The fact that macaque face selective regions respond to illusory faces[35] indicates that these stimuli can be used to study general face processing mechanisms at the neural level, without assuming that macaques experience the illusion as humans do. Moreover, since these stimuli activate face-selective systems without the confounding presence of recognizable face parts, they offer unique insights into holistic processing in face cells.

Here, we leverage face pareidolia images to address whether face cell responses to illusory faces mirror the holistic perception observed in humans and possibly monkeys. We recorded multiunit activity in central (CIT) and anterior IT (AIT) in 8 monkeys ($n = 4$ for CIT, $n = 5$ for AIT). Specifically, we investigate 1) whether macaque face cells selectively respond to images with an illusory face compared to those without, 2) if face-cell responses reflect how face-like an image looks to a human observer, 3) if a face-like configuration is necessary for driving neural pareidolia selectivity, and 4) what other, non-holistic attributes drive pareidolia selectivity. Additionally, we collected human behavioral data to investigate the contribution of the face-like configuration and other non-holistic attributes in perceptual pareidolia. Overall, we found that face-selective units in IT were pareidolia selective, responding more to pareidolia images than to matched controls. Moreover, face selectivity and pareidolia selectivity were positively correlated in both central and anterior IT. However, human faceness ratings did not explain face-cell activity to pareidolia images, and a linear combination of the neural responses only weakly predicted human faceness ratings. Eliminating the face-like configuration did not abolish pareidolia selectivity of face cells, or the correlation between face and pareidolia selectivity. When we presented the four quadrants in isolation, we found that the pareidolia selectivity of face-selective units was driven primarily by the quadrants containing the object features representing the eyes of the illusory face. Further, we found that pareidolia selectivity was predicted by neural tuning estimated from non-face, non-pareidolia images. Finally, in contrast to pareidolia selectivity of face cells, we found that quadrant scrambling significantly impacted how easily human subjects perceived a face in pareidolia images and that perceptual pareidolia was not driven primarily by pareidolic eye features. Taken together, we show that individual face-cell responses to pareidolic images are mostly driven by non-holistic object features, unlike human perceptual pareidolia, which requires the holistic face-like configuration.

## Results

In our main experiment, we presented 100 pareidolia images and 100 control images to 8 macaque monkeys in a Rapid Serial Visual Presentation (rsvp)-style paradigm. The pareidolia images were images of objects that evoked the illusion of a face, whereas control images were matched in object identity to the pareidolia images but did not evoke an illusory face experience[31,34]. We recorded neural activity from central (CIT) and anterior IT (AIT) while these images were presented. To investigate the relationship between pareidolia selectivity of face cells and human behavioral ratings, we presented the monkeys with a new set of 200 pareidolia images that had been given faceness ratings on an 11-point scale (0 "cannot see a face" to 10 "easily see a face") by human subjects in a recent study[34]. To investigate the contribution of the face-like configuration on the pareidolia selectivity of face cells, we presented quadrant-scrambled pareidolia and control images, in which each image was divided into four quadrants and the quadrants shuffled. This ensured that the individual features were preserved but the images did not retain the global configuration and thus did not evoke the perceptual experience of a face. To further investigate the contribution of non-holistic attributes in the pareidolia selectivity, we presented the four quadrants of the original and scrambled configurations in isolation. Finally, we also collected human faceness ratings on an 11-point scale (0 "cannot see a face" to 10 "easily see a face") for quadrant-scrambled images and isolated quadrants to investigate the contribution of the face-like configuration and other non-holistic attributes in human perceptual pareidolia.

### Units in CIT and AIT show pareidolia selectivity

In the first experiment, we recorded multiunit spiking activity in CIT ($n = 208$ sites, pooled across 4 macaques) and AIT ($n = 163$ sites, pooled across 5 macaques). Two of the macaques included in this study had participated in a previous study investigating the effect of withholding face experience during early development. However, at the time the current experiments were run, both monkeys had more than two years' experience with faces. We used 40 faces, 40 non-faces, 100 pareidolia images and 100 matched control objects (see examples in Fig. 1a). We found that, on average, the recorded populations contained a range of face selective (responded more to faces than nonfaces) and pareidolia selective (responded more to pareidolia images than to matched controls) units in CIT (Fig. 1b, c) and AIT (Fig. 1f, g). In Fig. 1d, e, h, i, we show the time course of the response for example face and nonface units. Next, we investigated whether the magnitude of a unit's face selectivity predicted the magnitude of its pareidolia selectivity.

### Pareidolia selectivity is correlated with face selectivity

We quantified category selectivity for each unit by computing a d' selectivity index for faces and a d' for pareidolias. This index reflects the difference in mean responses between two image categories in standard deviation units - for instance, a face $d' > 0$ indicates units respond more to faces (Fig. 1a, red rectangle) than to non-faces (Fig. 1a, gray rectangle); pareidolia $d' > 0$ indicates units respond more to pareidolia (Fig. 1a, orange rectangle) than to matched controls (Fig. 1a, blue rectangle). Further, we use the term "face units" to refer to units with a face $d' > 1$ and "nonface units" to refer to those with a face $d' < 1$. The average pareidolia d' for face units was $0.3 \pm 0.16$ ($t_{112} = 19.36$, $p = 1.6 \times 10^{-37}$) whereas for non-face units it was $0.04 \pm 0.16$ ($t_{94} = 2.3$, $p = 0.03$) in CIT. In AIT, the average pareidolia d' for face units was $0.53 \pm 0.32$ ($t_{121} = 17.98$, $p = 5.52 \times 10^{-36}$), whereas for non-face units it was $0.3 \pm 0.2$ ($t_{40} = 9.2$, $p = 2.01 \times 10^{-11}$). These averages are lower than the threshold face d' value of 1 used to select face-selective units, which is expected given the overall lower response to pareidolia images (see Fig. 1). In both regions, the average pareidolia d' was significantly higher for face units than for non-face units (CIT: $t_{206} = 11.36$,

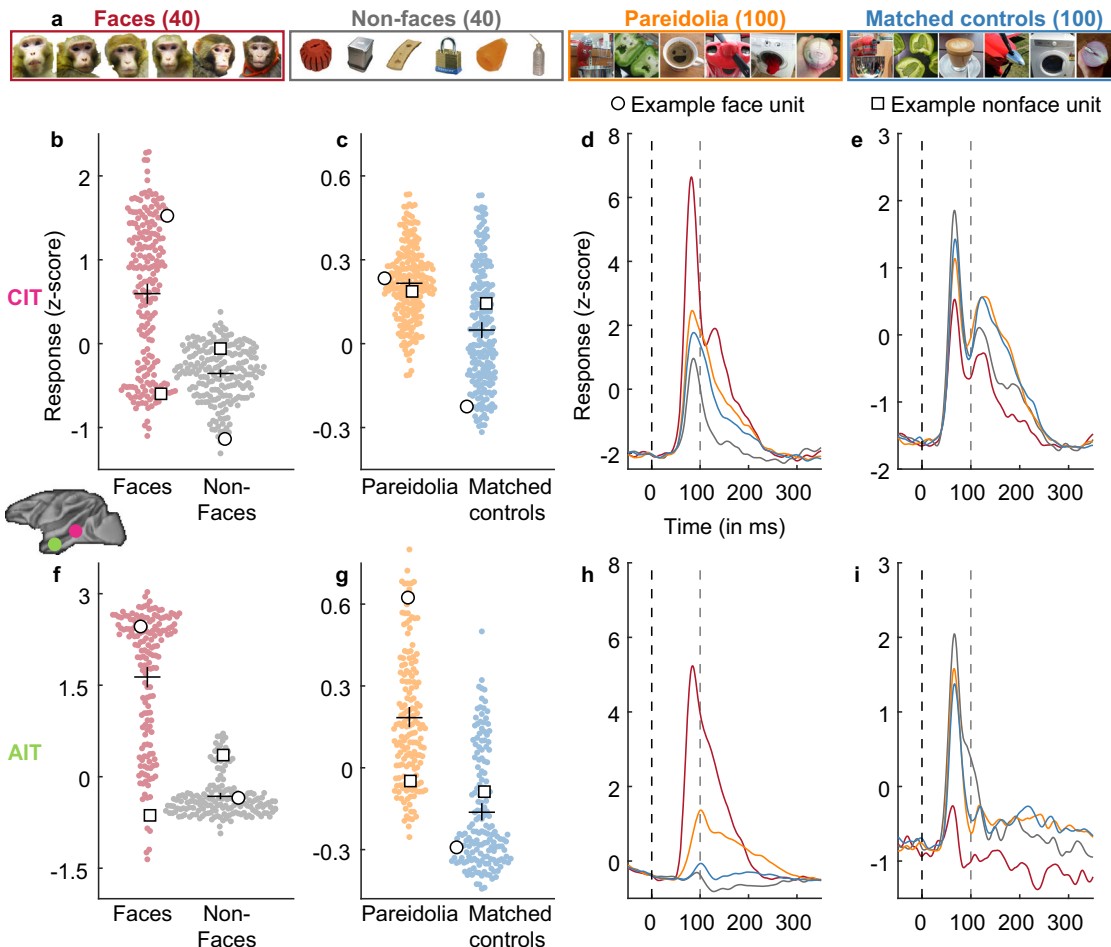

**Fig. 1 | Units in CIT and AIT show pareidolia selectivity. a** Example visual stimuli used in the experiment. Faces (red box) and non-face objects (gray box) were included to be able to calculate the face selectivity of the neural units. Each pareidolia image (orange box) included a matched control object (blue box), which had the same object identity, but did not evoke the perceptual experience of a face. Pareidolia and matched control image examples adapted from Wardle, S.G., Taubert, J., Teichmann, L. et al. Rapid and dynamic processing of face pareidolia in the human brain. Nat Commun 11, 4518 (2020). https://doi.org/10.1038/s41467-020-18325-8 released under a CC BY license: https://creativecommons.org/licenses/by/4.0/. **b** The beeswarm plot shows normalized neural response per unit averaged across 150 ms time window starting at response onset (see Methods) to faces (red) and non-faces (gray) in central IT (CIT; $n = 208$, pooled across 4 monkeys). The black central horizontal line shows the mean response, the black central vertical line indicates confidence intervals. Open circles and squares represent the example face (circle) and nonface (square) units shown in (**d, e. c**). Plot showing normalized neural response per unit to pareidolia (orange) and matched control objects (blue). Same conventions as in (**b**). **d** The time course of an example face unit (depicted by an open circle in b, c) in CIT. X-axis represents time in milliseconds and y-axis shows the normalized neural response. Vertical dashed line at t = 0 indicates stimulus onset. Gray line at $t = 100$ indicates stimulus offset. **e** Time course of an example nonface unit (depicted by an open square in **b, c**). Same conventions as in (**d**). **f, g** Beeswarm plots showing normalized neural responses per unit anterior IT (AIT; $n = 163$, pooled across 5 monkeys). Same conventions as in (**b, c**). Time course of an example face unit (**h**) and nonface unit (**i**) in AIT. Same conventions as in (**d, e**). The small inset of the macaque brain shows the approximate location of the recording sites in CIT (pink circle) and AIT (green circle).

$p = 1.56 \times 10^{-23}$, AIT: $t_{161} = 4.38$, $p = 2.15 \times 10^{-5}$). The pareidolia and face selectivity of each unit are shown in Fig. 2a, b. Pareidolia selectivity was positively correlated with face selectivity (Fig. 2a, b; CIT: $n = 208$, Pearson's $r = 0.7$, $p = 9.39 \times 10^{-32}$, AIT: $n = 163$, Pearson's $r = 0.61$, $p = 7.62 \times 10^{-18}$). This positive correlation between face and pareidolia selectivity was present in each of the 8 monkeys individually (Supplementary Table 1). One-sample t-tests on the Fischer-transformed correlation values indicate that this correlation was significant across monkeys (CIT: $t_3 = 4.62$, $p = 0.0096$; AIT: $t_4 = 8.19$, $p = 0.00061$). This means that a unit's face selectivity and its pareidolia selectivity are closely related in IT cortex and could both be driven by image attributes shared between faces and pareidolia images. If both the human perceptual illusion of face pareidolia and pareidolia selectivity of face cells are driven by the resemblance of an object to an actual face, then our findings raise the question: are face-cells activated by the same features that make an object looks like a face to a human observer?

## Faceness does not explain pareidolia selectivity in face cells

To explore the association between visual perception and neural response magnitude, we investigated how the neural responses of face cells to pareidolia images relate to human "faceness" ratings of the same images. To this end, we used a set of 200 pareidolia images and matched controls, which human subjects had rated on an 11-point scale from 0 "cannot see a face" to 10 "easily see a face" in a recent study[34]. Interestingly, not all pareidolic images had received high faceness ratings, leading to similar ratings for some pareidolic and the most "face-like" control images. If individual face-cells encode how much an object looks like a face, then face-cell responses should not differ for images that received the same faceness rating. To test this, we selected the ten least face-like pareidolic images (faceness rating: $2.9 \pm 0.8$) and the 10 most face-like control images (faceness rating: $3 \pm 0.5$), which were not necessarily matched in identity (Fig. 3a, b). Next, we computed the pareidolia d' using these faceness-matched pareidolia and control images. We found that face units (face d' > 1) responded more

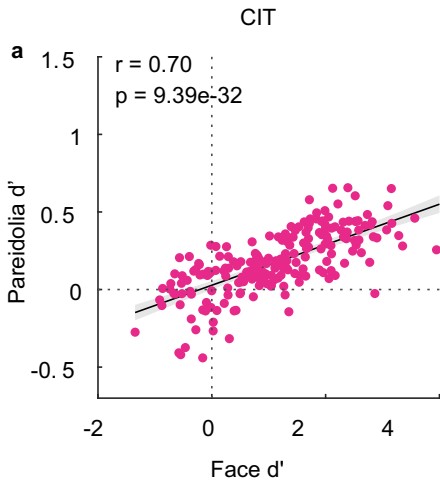

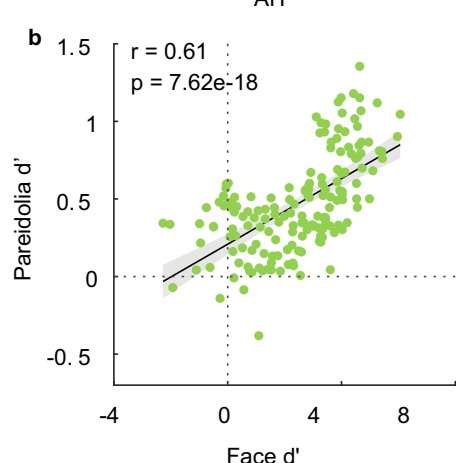

**Fig. 2 | Pareidolia selectivity is correlated with face selectivity. a** Scatterplot showing the correlation between face selectivity (face $d'$) plotted on the x-axis and pareidolia selectivity (pareidolia $d'$) on the y-axis. Each dot depicts a neural unit in central IT (CIT; $n = 208$, pooled across 4 monkeys). The black line indicates an ordinary least squares (OLS) linear regression fit, with shaded 95% confidence intervals error bands. The values on the top left corner depict the Pearson's correlation ($r$) and the corresponding $p$-value calculated using the corr function in MATLAB (R2021b). **b** Scatterplot showing the correlation between face and pareidolia selectivity in anterior IT (AIT; $n = 163$, pooled across 5 monkeys). Same conventions as in (**a**).

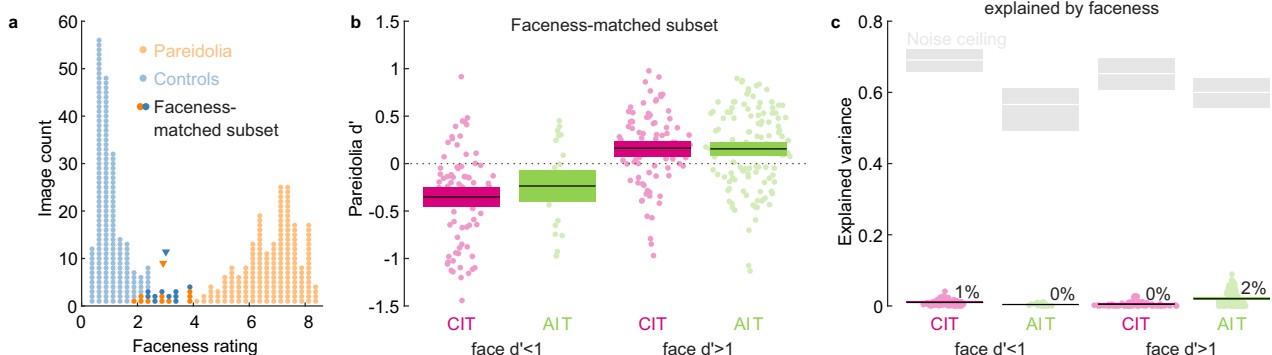

**Fig. 3 | Faceness does not explain pareidolia selectivity in face cells.**
**a** Distribution of perceptual faceness ratings for all 200 pareidolia images (orange dots) and their matched controls (blue dots) taken from Wardle et al. 2022. Darkened dots indicate a faceness-matched subset of the 10 most face-like control images and the 10 least face-like pareidolia images. Downward-pointing triangles indicate the means of the faceness-matched subsets. **b** Pareidolia d′ values computed using the 20 faceness-matched image subset for nonface (face $d'<1$, CIT = 84, AIT = 24) and face units (face $d'>1$, CIT = 87, AIT = 120). Each marker represents a single neural site. Colored boxes: 95% bootstrapped CI, calculated by resampling sites; black line: mean. **c** Response variance for all 200 pareidolia images explained by faceness ratings. Black line: mean. Colored boxes indicating 95% bootstrapped CI are obscured by the black mean line. Gray boxes and the white line indicate the 95% bootstrapped CI and mean across neural sites of the noise ceiling (i.e., the maximum explainable variance, computed as the neural response reliability multiplied by the reliability of the faceness-rating). *CIT* central IT, *AIT* anterior IT.

to illusory faces than to controls (pareidolia $d'$, CIT: 0.16, $t_{86} = 3.8$, $p = 0.0002$; AIT: 0.15, $t_{119} = 4.1$, $p = 8 \times 10^{-5}$; Fig. 3c), even though there was no difference in average faceness ratings. While it is difficult to interpret these results, they demonstrate that pareidolia selectivity can persist even in the absence of a difference in faceness ratings. Presumably, some pareidolia images selected by the authors turned out to not be very face-like, despite sharing one or more features that face cells respond to.

To further investigate the relation between neural responses and faceness ratings, we asked whether the response differences between individual pareidolia images could be explained by differences in faceness. For this analysis, we used all 200 pareidolia images. We computed for each unit the variance in pareidolia responses explained by faceness ratings (squared Pearson correlation between responses and faceness ratings). For face units, the faceness ratings explained on average ~0% (95%CI[0.3 0.6]) of the pareidolia response variance in CIT, and ~2% (95%CI[1.7 2.5]) in AIT (Fig. 3d). For units with face $d'<1$, the explained variance was ~1% (95%CI[0.8 1.2]) in CIT and ~0% (95%CI[0.3 0.6]) in AIT. Thus, faceness ratings did not explain response variance for pareidolia images. Even if responses of individual neural sites do not encode the perceptual faceness of pareidolia images, it could still be that the perceptual faceness can be linearly decoded from the neural population. To test this, we fit a cross-validated linear regression (see Methods) predicting faceness ratings from the population response pattern. For both CIT and AIT, there was a weak correlation between observed and predicted faceness (CIT: $r_{198} = 0.16$, $p = 0.03$,

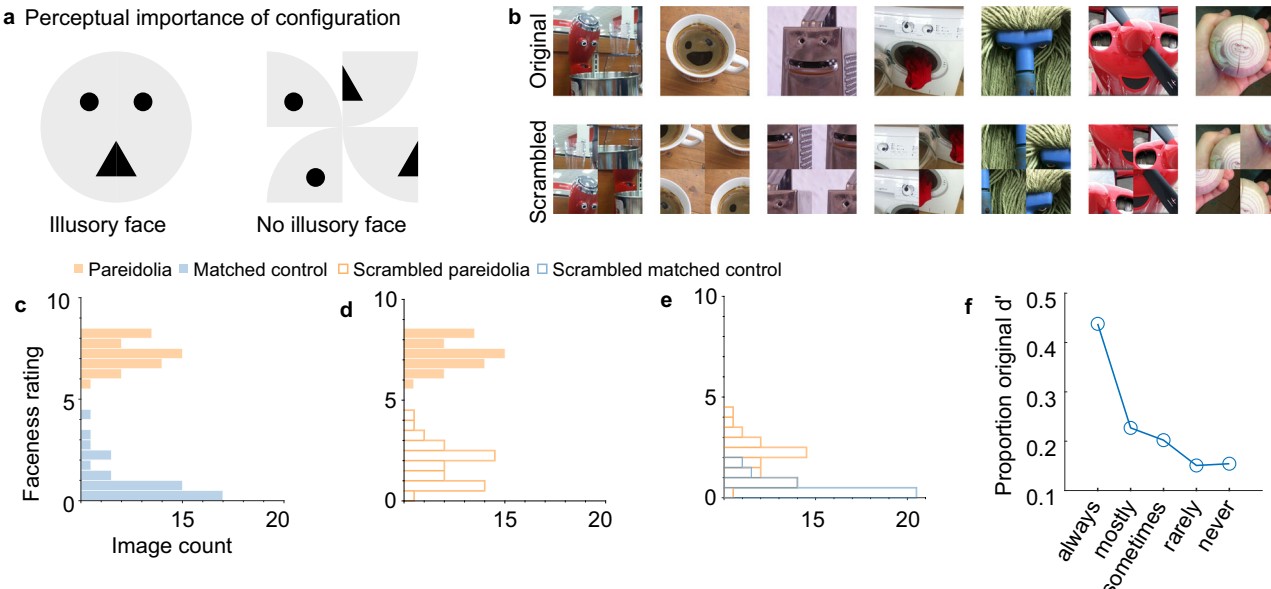

**Fig. 4 | Quadrant scrambling disrupts the faceness of pareidolia images.**
**a** Cartoon example demonstrating the importance of configuration to evoke the perceptual experience of a face. **b** Example pareidolia images with their original (top) and quadrant scrambled versions (bottom), adapted from Wardle, S.G., Taubert, J., Teichmann, L. et al. Rapid and dynamic processing of face pareidolia in the human brain. Nat Commun 11, 4518 (2020). https://doi.org/10.1038/s41467-020-18325-8 released under a CC BY license: https://creativecommons.org/licenses/by/4.0/. We divided each image into four equal quadrants and then shuffled the location of each quadrant, so that the individual features were maintained, but no longer in their original face-like configuration. **c−e** Distribution of perceptual faceness ratings collected from human subjects for a subset of 34 original pareidolia images (filled orange bars), matched control objects (filled blue bars), scrambled pareidolia images (open orange bars), scrambled matched controls (open blue bars). **f** Line plot depicting the faceness rating $d'$ for the scrambled images (normalized by the rating $d'$ for the original images) as a function of the subjects' response to the post-task question "Some of the images that you saw appeared jumbled. How frequently did you try to mentally shift the parts to make them whole again?", with the following answer choices as a response: "Never, rarely, sometimes, most of the time, always". Out of 100 subjects, 11 responded "never", 15 responded "rarely", 29 responded "sometimes", 30 responded "most of the time", and 15 responded "always".

95%CI[0.02 0.29]; AIT: $r_{198} = 0.26$, $p = 0.0002$, 95%CI[0.12 0.38]). The square of these correlations is 2.5% for CIT (noise ceiling 61.4%) and 6.7% for AIT (noise ceiling 78.1%), which is only slightly higher than the average explained variance for individual neural sites (-0−2%). Taken together, these findings further support the notion that pareidolia selectivity of face cells is not explained by the "faceness" of the pareidolia images as perceived by humans. Thus, our results indicate a discrepancy between the cues for the human perception of faceness and those that activate face cells. Next, we investigated to what extent the global, face-like configuration is a critical cue for the pareidolia selectivity of face cells as well as the human perceptual experience of an illusory face.

**Quadrant scrambling disrupts faceness of pareidolia images**
A critical factor behind the illusory perception of a face may be the global configuration or the spatial arrangement of certain object parts, such as circular spots or straight lines. When arranged in the T-shaped arrangement typically associated with faces, such object parts can appear as two eyes above a mouth, thus evoking the perceptual experience of an illusory face (see examples in Fig. 4b, top). This face-like global configuration has been suggested to facilitate rapid face detection through template matching, even in the absence of normal facial features[11,41,43–46]. Further, the preference for the face-like configuration over a scrambled configuration, even when composed of simple shapes such as circles or squares (such as in our cartoon example in Fig. 4a), is already present in early infancy and continues into adulthood for both humans and monkeys[47–55]. Moreover, the face-like configuration, but not the scrambled configuration, has been shown to interfere with local feature detection or peripheral face recognition, suggesting that we find such a configuration difficult to ignore[56,57]. Collectively, extensive prior research underscores the

significance of the global face-like configuration in face perception. Indeed, this significance becomes apparent in the cartoon example in Fig. 4a: the perception of a face is strong when two dots are horizontally aligned and above a high-contrast triangle. Yet, the illusion of a face is eliminated when the face-like configuration is abolished by scrambling the configuration (See cartoon example in Fig. 4a and scrambled pareidolia examples in Fig. 4b, bottom row).

To test the importance of configuration for human pareidolia perception, we collected human faceness ratings for a subset of 34 pareidolia images and matched control objects, plus quadrant-scrambled versions of these images, created by dividing each image into four quadrants, then shuffling the quadrants, so the new image retained the individual features of the pareidolia image without the original face-like spatial configuration (Fig. 4b). Subjects indicated on an 11-point scale how easily they could see a face in the presented images, with 0 indicating "cannot see a face" and 10 indicating "easily see a face". To ensure that the ratings were not driven by image familiarity, no subject saw the scrambled version of an original image they had seen and vice versa (see Methods for details). Furthermore, upon completion of the experiment, we asked subjects to what extent they had mentally shifted the parts of scrambled images to make them whole again. The original pareidolia images received high faceness ratings (Fig. 4c, mean rating = 7.23 ± 0.76) compared to matched control images (Fig. 4c, mean = 0.94 ± 1.01), demonstrating that subjects could more easily see a face in the pareidolia images than in the matched controls ($t_{33} = 32.01$, $p = 1.99 \times 10^{-26}$). Moreover, subjects rated the original pareidolia images significantly higher than the scrambled images (Fig. 4d, original = 7.23 ± 0.76, scrambled = 1.84 ± 0.94, $t_{33} = 33.9$, $p = 3.27 \times 10^{-27}$), emphasizing the importance of global configuration in perceptual pareidolia. The scrambled pareidolia images (mean = 1.84 ± 0.94) were rated higher

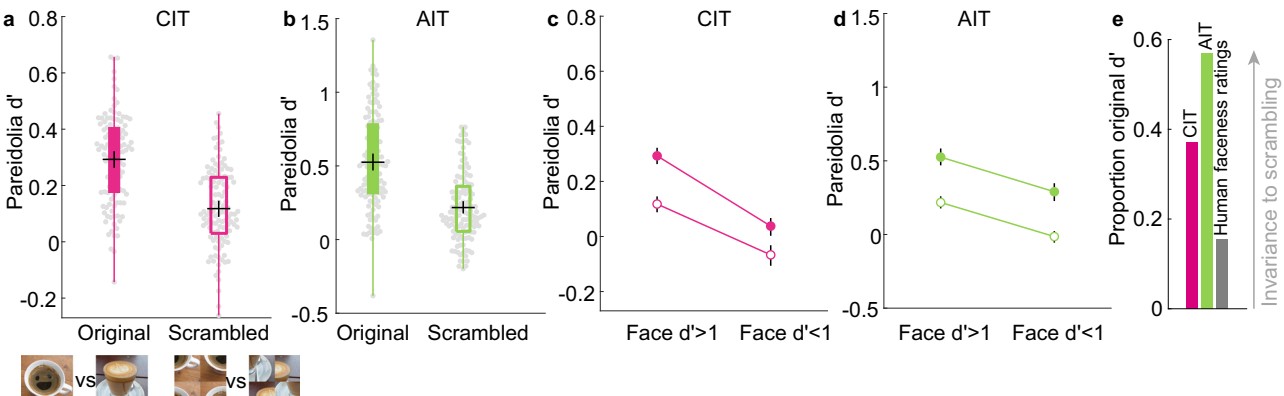

**Fig. 5 | Quadrant scrambling does not eliminate neural pareidolia selectivity.**
**a** Average pareidolia selectivity for 100 scrambled pareidolia images vs 100 scrambled matched controls is shown in box plots for macaque face units (face $d' > 1$) in central IT (CIT; $n = 113/208$) for original (filled boxplot) and scrambled (open boxplot) images. The black central horizontal line shows the mean response, the black central vertical line indicates confidence intervals, and the bottom and top edges of the box depict the 25th and 75th percentiles, respectively. Whiskers (vertical lines in color) show the most extreme data points not considered outliers. Beeswarm plots behind the box plots show the pareidolia selectivity of each individual unit as a gray dot. **b** Average pareidolia selectivity for face units in anterior IT (AIT; $n = 122/163$). Same conventions as in (**a**). **c** Plot showing the average pareidolia selectivity of face (face $d' > 1$; $n = 113$) and nonface (face $d' < 1$; $n = 95$) units in CIT for original (filled) and scrambled (open) pareidolia images. Error bars indicate confidence intervals. **d** Plot showing average pareidolia selectivity for face (face $d > 1$; $n = 122$) and nonface (face $d' < 1$; $n = 41$) units in AIT. Same conventions as in (**c**). **e** The scrambled pareidolia $d'$ (normalized by the rating d' for the original images) for CIT (pink bar), AIT (green bar), and human faceness ratings (gray bar; averaged across the subjects in the "never" group, see Fig. 4f) for a subset of 34 images. Example pareidolia and matched control images shown in the legend of (**a**). adapted from Wardle, S.G., Taubert, J., Teichmann, L. et al. Rapid and dynamic processing of face pareidolia in the human brain. Nat Commun 11, 4518 (2020). https://doi.org/10.1038/s41467-020-18325-8 released under a CC BY license: https://creativecommons.org/licenses/by/4.0/.

than the scrambled matched controls (0.50 ± 0.45), suggesting that scrambled pareidolia images were still perceived as somewhat more face-like (Fig. 4e, $t_{33} = 7.88$, $p = 4.34 \times 10^{-9}$). However, the difference in ratings between scrambled pareidolia and scrambled matched controls decreased as a function of how often the subjects reported having unscrambled images ('always' group, difference in rating = 2.23 ± 0.97; "mostly" = 1.67 ± 0.87; "sometimes" = 1.16 ± 0.62, "rarely" = 0.74 ± 0.85, 'never' = 0.55 ± 0.51). Thus, mental unscrambling can explain how easily a face was perceived in the scrambled pareidolia compared to scrambled matched controls (Fig. 4f). Furthermore, for subjects in the 'never' group, scrambling the pareidolia images lowered perceived faceness to the level of matched control objects (scrambled pareidolia = 0.91 ± 0.87, original matched control = 0.65 ± 1.33, $t_{27} = 0.84$, $p = 0.41$). This result provides further evidence that perceptual pareidolia is driven by the global configuration, since only subjects who reported that they mentally unscrambled the scrambled pareidolia images showed substantial pareidolia effects.

## Quadrant scrambling does not eliminate neural pareidolia selectivity

Next, we investigated the impact of quadrant scrambling on pareidolia selectivity of face cells. We computed the scrambled pareidolia selectivity of face cells using 100 scrambled pareidolia images vs 100 scrambled matched control objects (from Experiment 1). We used scrambled rather than original matched control images to compute scrambled pareidolia selectivity since scrambling has an overall effect on response strength, regardless of stimulus condition. Quadrant scrambling of the pareidolia images did not eliminate pareidolia selectivity of face units (Fig. 5a, b; CIT = 0.12 ± 0.16, $t_{112} = 8.04$, $p = 1.05 \times 10^{-12}$, AIT = 0.22 ± 0.23, $t_{121} = 10.3$, $p = 3 \times 10^{-18}$). The positive correlation between face and pareidolia selectivity was also preserved (CIT: $n = 208$, Pearson's $r = 0.5$, $p = 2.49 \times 10^{-14}$, AIT: $n = 163$, Pearson's $r = 0.62$, $p = 7.95 \times 10^{-19}$) and significant across monkeys (Supplementary Table 1; CIT: $t_3 = 2.4$, $p = 0.049$; AIT: $t_4 = 4.61$, $p = 0.005$). Nevertheless, scrambling the global configuration of the pareidolia images did cause a significant reduction in the average pareidolia selectivity of

face cells. Therefore, we asked if scrambling the global configuration impacted nonface units as well. If some of the pareidolia selectivity of face cells is driven specifically by the facial configuration, the impact of scrambling should be restricted to face cells (i.e., neurons that could potentially respond selectively to a facial configuration). In Fig. 5c, d, we compare pareidolia selectivity of face and nonface cells for original vs scrambled images. In CIT, scrambling the global configuration significantly reduced the average pareidolia selectivity for face units (original = 0.29, scrambled = 0.12, $t_{112} = 9.98$, $p = 3.55 \times 10^{-17}$). However, scrambling the global configuration also significantly reduced the pareidolia selectivity of nonface units (original = 0.04, scrambled = −0.07, $t_{94} = 9.04$, $p = 2.02 \times 10^{-14}$). Likewise, in AIT, scrambling significantly reduced the average pareidolia selectivity for face (original = 0.53, scrambled = 0.22, $t_{121} = 16.5$, $p = 9.32 \times 10^{-33}$) and nonface units (original = 0.3, scrambled = −0.01, $t_{40} = 13.7$, $p = 1.002 \times 10^{-16}$). A threshold of face $d' < 1$ for nonface units could still include units weakly selective for faces. However, even when we define the nonface units as face $d' < 0$ (indicating that they show stronger responses to nonfaces than faces), the significant reduction in selectivity as an impact of scrambling remained (CIT: original = −0.025, scrambled = −0.0995, $t_{39} = 4.65$, $p = 3.71 \times 10^{-5}$; AIT: original = 0.3, scrambled = −0.03, $t_{15} = 10.32$, $p = 3.29 \times 10^{-8}$).

While scrambling did result in a general reduction in pareidolia selectivity for both face and nonface units, the effect of scrambling was significantly larger for the face units than for the nonface units in CIT (Fig. 5c, $t_{206} = 3.23$, $p = 0.0015$). Indeed, this is reflected in the significant reduction by scrambling in the correlation between face selectivity and pareidolia selectivity across all CIT units (original $r = 0.7$, scrambled $r = 0.5$, difference in $r = -0.2$, 95%CI [−0.28, −0.11]). In AIT, however, the effect of scrambling on face units was not different from the effect of scrambling on nonface units (Fig. 5d, $t_{161} = 0.098$, $p = 0.92$). Correspondingly, there was no difference in the correlation between face and pareidolia selectivity across all AIT units (original $r = 0.61$, scrambled $r = 0.62$, difference in $r = 0.01$, 95%CI [−0.05, 0.08]). Thus, our results suggest that the effect of scrambling on pareidolia selectivity was independent of face selectivity, though more so in AIT than in CIT. We provide in the discussion a potential explanation as to

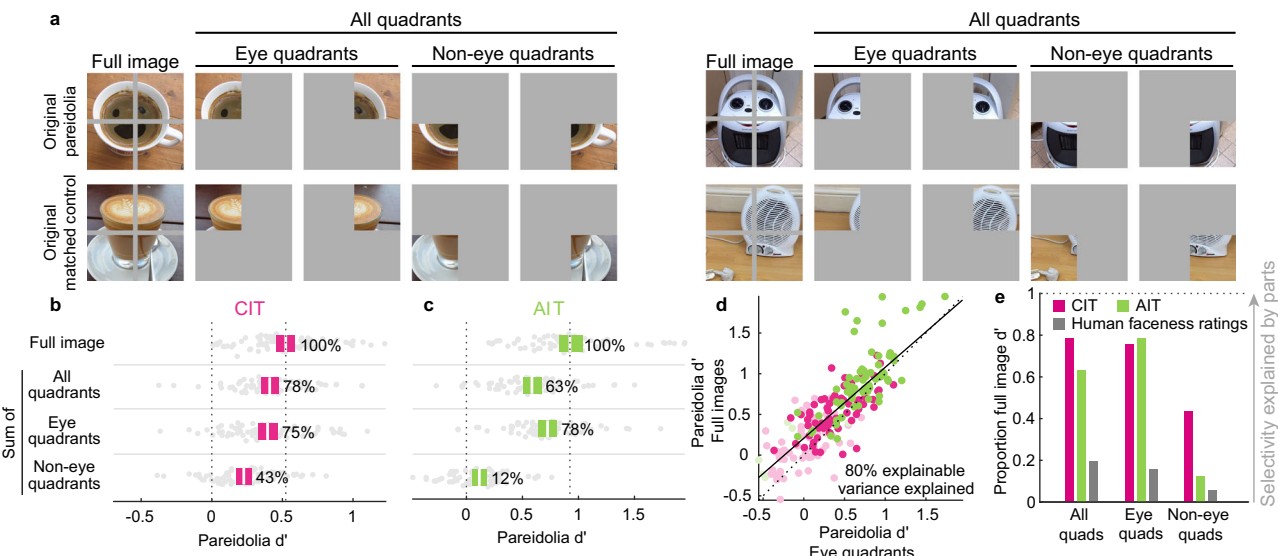

Fig. 6 | **Neural pareidolia selectivity in face units is primarily driven by "eye" quadrants. a** Two example pareidolia and the corresponding matched control images, and the isolated quadrants. A gray cross was introduced in the full original images to ensure the shared presence of quadrant edges in both original and scrambled images. The gray background in the single-quadrant images was the same color as the screen background. The quadrants were divided into "eye" and "non-eye" categories. **b** The pareidolia d′ in central IT (CIT) for face units (face d′ > 1; n = 69) for full images, the average d′ for the sum of all quadrants, the average d′ for the sum of only the eye quadrants and the average d′ for the sum of only the non-eye quadrants for original images. Colored boxes indicate 95% bootstrapped CI, calculated by resampling sites, with the central vertical line indicating the mean d′ across units. The numbers adjacent to the colored boxes indicate the percentage of the mean d′ value relative to the d′ of the full original images. Beeswarm charts show

the d′ of each neural unit, where each gray circle represents a neural unit. **c** Same conventions as in (**b**), except for face units in anterior IT (AIT; n = 65). **d** Scatter plot showing distribution of pareidolia d′ for the eye quadrants only from original images on the x-axis, and the pareidolia d′ for the full original image on the y-axis. Each circle represents a neural unit in CIT (pink) and AIT (green). Lighter circles are face cells with face d′ < 1 and darker circles represent units with face d′ > 1. **e** The proportion of mean d′ value (relative to the d′ of the full original images) for all quadrants, eye quadrants only and non-eye quadrants only for CIT (pink bar), AIT (green bar), and human faceness ratings (gray bar). Example images shown in (**a**). adapted from Wardle, S.G., Taubert, J., Teichmann, L. et al. Rapid and dynamic processing of face pareidolia in the human brain. Nat Commun 11, 4518 (2020). https://doi.org/10.1038/s41467-020-18325-8 released under a CC BY license: https://creativecommons.org/licenses/by/4.0/.

why nonface units in CIT were less affected by scrambling, unlike in AIT.

So far, our results indicate that quadrant scrambling reduces but does not eliminate either human perceptual pareidolia or pareidolia selectivity of face cells. How does the reduction in perceptual pareidolia and neural pareidolia selectivity compare with each other? We computed the scrambled pareidolia selectivity in CIT and AIT using the same 34 images that the human subjects saw (Fig. 5e). The human faceness rating d′ of the quadrant scrambled images only amounted to 15.4% of the original d′, whereas scrambled pareidolia d′ amounted to 37% for CIT and 57% for AIT of the original d′. Overall, these results show that quadrant scrambling reduced but did not eliminate the pareidolia selectivity in CIT and AIT, as opposed to the much larger effect on human perceptual pareidolia. Further, the reduction in neural pareidolia selectivity was largely independent of face selectivity, suggesting that scrambling the global configuration affects IT neurons' responses generally, independently of their face selectivity and thus independent of their potential selectivity for a facial configuration.

### Neural pareidolia selectivity in face units is primarily driven by "eye" quadrants

Although quadrant scrambling did not abolish the pareidolia selectivity of face cells, it did reduce pareidolia selectivity. Was this reduction because scrambling disrupts the spatial configuration of the object parts or because it changes the absolute position of individual parts? To answer this question, we took the top 50 pareidolia images from Experiment 1 based on neural responses and their matched control objects, and presented the full original or scrambled versions, but also the four quadrants in isolation in the original and scrambled positions (Fig. 6a, Supplementary Fig. 2). Since quadrant scrambling introduces edges normally absent in the original images (see Fig. 4a for

examples), in this experiment we introduced a gray cross in both sets of full images. For clarity, when discussing the results of this experiment in this section, we report the average pareidolia d′ of face units (face d′ > 1) for the different image manipulations as a percentage of the average pareidolia d′ based on the full original image. Further, to compute the pareidolia d′ based on the individual quadrants, we used the individual quadrant of the pareidolia images (Fig. 6a, Supplementary Fig. 2a, upper row) and the corresponding quadrant of the matched control objects at the same location (Fig. 6a, Supplementary Fig. 2a, lower row). We compared pareidolia d′ rather than absolute response magnitudes, because presenting scrambled images or isolated quadrants (i.e., smaller images) could affect the overall response magnitude for all images, without necessarily affecting the pareidolia selectivity. Finally, we included only 34/50 pareidolia images in our analysis, where the eyes and mouth were clearly separable and confined to different quadrants.

To isolate the effect of configuration from the effect of position, we looked at the average pareidolia d′ based on the sum of the responses to the individual four image quadrants (of pareidolia images and controls) presented in their original positions (Fig. 6b, c). The average pareidolia selectivity based on the summed response of these four quadrants was 78% in CIT and 63% in AIT. Thus, removing only the global configuration (face-like or otherwise), while retaining the position of the object parts, reduced pareidolia selectivity by only 22% (100%–78%; $t_{68} = 3.7$, $p = 0.00046$) in CIT and by 37% in AIT (100%–63%; $t_{64} = 9.78$, $p = 2.57 \times 10^{-14}$). Next, we categorized the quadrants into "eye" and "non-eye" quadrants, depending on which part of the illusory face they represented in the original pareidolia image, to investigate their individual contribution to pareidolia selectivity. The average pareidolia selectivity for the sum of the eye quadrants was 75% of the average for the full original images, which is not

significantly different from the selectivity based on the sum of all quadrants (78%–75%; $t_{68} = 0.75$, $p = 0.46$). Furthermore, pareidolia selectivity based on the sum of individual eye quadrants in the scrambled location was 67% (Supplementary Fig. 2a), which is not significantly different from the selectivity for the sum of individual eye quadrants in their original location (75%–67%; $t_{68} = 1.4$, $p = 0.18$). Finally, there was still significant pareidolia selectivity for the sum of non-eye quadrants in the original location (43%), but not in the scrambled configuration (−6%, Supplementary Fig. 2a). Overall, these results suggest that pareidolia selectivity in CIT is driven mostly by eye parts, irrespective of their position, and reduced by the scrambled positions of the non-eye parts. In AIT, the average pareidolia selectivity based on only the sum of eye quadrants was even closer to the average for the full original images (78% and 81% for original and scrambled positions, respectively). Moreover, pareidolia selectivity for the non-eye quadrants was negligible (12–15%) compared to the magnitude for the full original images. Taken together, our results indicate that—like central IT—pareidolia selectivity in anterior IT is largely explained by eye quadrants, with a relatively smaller contribution of the configuration (face-like or otherwise). Indeed, this result can also be observed in Fig. 6d, where pareidolia selectivity based on eye quadrants explains 80% of the variance explained for the pareidolia selectivity of the full original images.

In contrast, human faceness ratings for the isolated eye quadrants were rated significantly lower than the full pareidolia images (eye quadrants = $1.1 \pm 0.82$, full pareidolia = $7.23 \pm 0.76$, $t_{33} = 20.98$, $p = 3.94 \times 10^{-42}$). Indeed, though the eye quadrants were rated marginally, albeit significantly higher than the matched control quadrants (eye quadrants = $1.1 \pm 0.82$, matched control quadrants = $0.37 \pm 0.47$; $t_{33} = 4.6$, $p = 6.1 \times 10^{-5}$), this amounted to only 15.5% of the full original $d'$ (Fig. 6e). This is in stark contrast to the neural data where the rating $d'$ for pareidolic eye vs matched control quadrants amounted to 75% for CIT and 78% for AIT of the full original $d'$ (Fig. 6e).

Thus, these results indicate that macaque neural pareidolia selectivity is primarily attributable to the pareidolic eye features, i.e., individual object parts that, when arranged in the right configuration, act as eyes of the illusory face. The face-like configuration itself appears to contribute minimally to pareidolia selectivity, less than the results with full scrambled versus full original images might suggest. In contrast, human perceptual pareidolia requires the global face-like configuration to evoke the perceptual illusion.

## Pareidolia selectivity can be captured by a non-face, non-pareidolia encoding model

Our results thus far show that pareidolia selectivity of face cells is primarily driven by individual object parts or local features and does not require an arrangement in a face-like configuration. This suggests that neural pareidolia selectivity results largely from a tuning for more generic features that apply to all kinds of objects, even ones that do not resemble a face. That is, certain generic features, like the dark holes in a leather belt or the roundness of an egg yolk, are likely more prevalent in images with illusory faces, but can also be present in the matched control objects. Does neural tuning for such individual features predict pareidolia selectivity?

In recent work, we showed that face selectivity can be predicted from features that apply to non-face objects[42]. Applying the same approach from our previous work, we quantified the tuning of each neural site to images without illusory or real faces and asked if this tuning could predict pareidolia selectivity in face cells. We computed the neural tuning by fitting an encoding model based on a convolutional neural network (CNN) trained on object categorization[58]. The model was pretrained on ImageNet, which does contain some faces, albeit not in a separate category. Critically, we used only matched controls (of the pareidolia images) or non-face object images to linearly map (fit) the features encoded by the CNN to the neural

responses (see "Methods" and Fig. 7a). We term this the non-pareidolia encoding model, because it characterizes the neural tuning using only responses to images that are not faces and do not contain illusory faces. Because an encoding model can be derived only from reliable responses, we excluded 105 sites (face $d'$ between −1.9 and 5.4) that had response reliability below the threshold of 0.4 for matched controls, leaving 266 remaining sites (face d′ between −2.3 and 6.1). This exclusion did not qualitatively affect the results. We then used the model-predicted responses to the 100 pareidolia images and matched controls used in Experiment 1 to compute a model-predicted pareidolia-selectivity value for each neural site. We found that the non-pareidolia encoding model accurately predicted the observed pareidolia selectivity (Fig. 7b; CIT: Pearson's r between observed and predicted values = 0.71, $p = 1.1 \times 10^{-27}$, AIT: Pearson's $r = 0.76$, $p = 2.5 \times 10^{-19}$). Furthermore, this model-predicted pareidolia selectivity correlated with the face selectivity observed in the actual neural responses (Fig. 7c; CIT: Pearson's $r = 0.68$, $p = 4.3 \times 10^{-24}$, AIT: Pearson's $r = 0.56$, $p = 4.5 \times 10^{-9}$). Hence, even though no pareidolia or face images were involved in quantifying the feature-tuning with the encoding model, the model successfully predicted both pareidolia selectivity and its relationship to the face selectivity of the actual neural sites.

What drives the pareidolia selective responses in the encoding model units? Like we did with the neural responses, we examined the encoding model's pareidolia selectivity to a subset of 34 scrambled images and to the isolated image quadrants (Fig. 7d). We found that, like macaque face-cells, the non-pareidolia encoding models of face-selective units (observed face $d' > 1$) are predominantly driven by eye parts. Finally, as a more stringent test of our hypothesis that generic object features predict pareidolia selectivity, we input both pareidolia and control images into the non-face encoding model of ref. 42. This non-face encoding model fit on an entirely independent dataset could still predict a degree of pareidolia selectivity that positively correlated with the face selectivity observed in the neural sites upon which the model was originally based (Supplementary Fig. 7a; Pearson's $r = 0.53$, $p = 3 \times 10^{-34}$). Furthermore, the model could also predict that the pareidolia selectivity of face units is driven primarily by eye parts only (Supplementary Fig. 7b). Overall, our results indicate that the link between pareidolia selectivity and face selectivity was captured by domain-general features, that apply to non-pareidolia, non-face images and were derived from a rich feature space that was trained to represent all kinds of objects.

## Discussion

In this study, we used images of illusory faces in inanimate objects to study holistic processing in face cells. Our aim was to test whether face selective units in the IT cortex respond to illusory faces because of the face-like configuration of individual object features, which is integral to the human perceptual experience of an illusory face. We found that face selective units (i.e., units that respond more strongly to faces than to nonfaces) in macaque central and anterior IT cortex were pareidolia selective (i.e., they responded more strongly to pareidolia images than to matched control objects). When investigating what drives this stronger response to pareidolia images, we found that 1) it could not be explained by human faceness ratings, 2) it did not require the global configuration that is critical for the perceptual experience of an illusory face, 3) it was instead primarily driven by local features in isolation, 4) it could be explained by generic features also present to varying degrees in non-pareidolia, nonface objects. The contribution of the global configuration to neural pareidolia selectivity was minor (up to 20–25%). In contrast, in humans, the perceptual illusion of a face depended primarily on the global configuration and not on local features. Taken together, our results indicate that neural pareidolia selectivity of macaque face cells is dissociable from the extent to which an object looks like a face to us. Instead, it can be explained mostly by

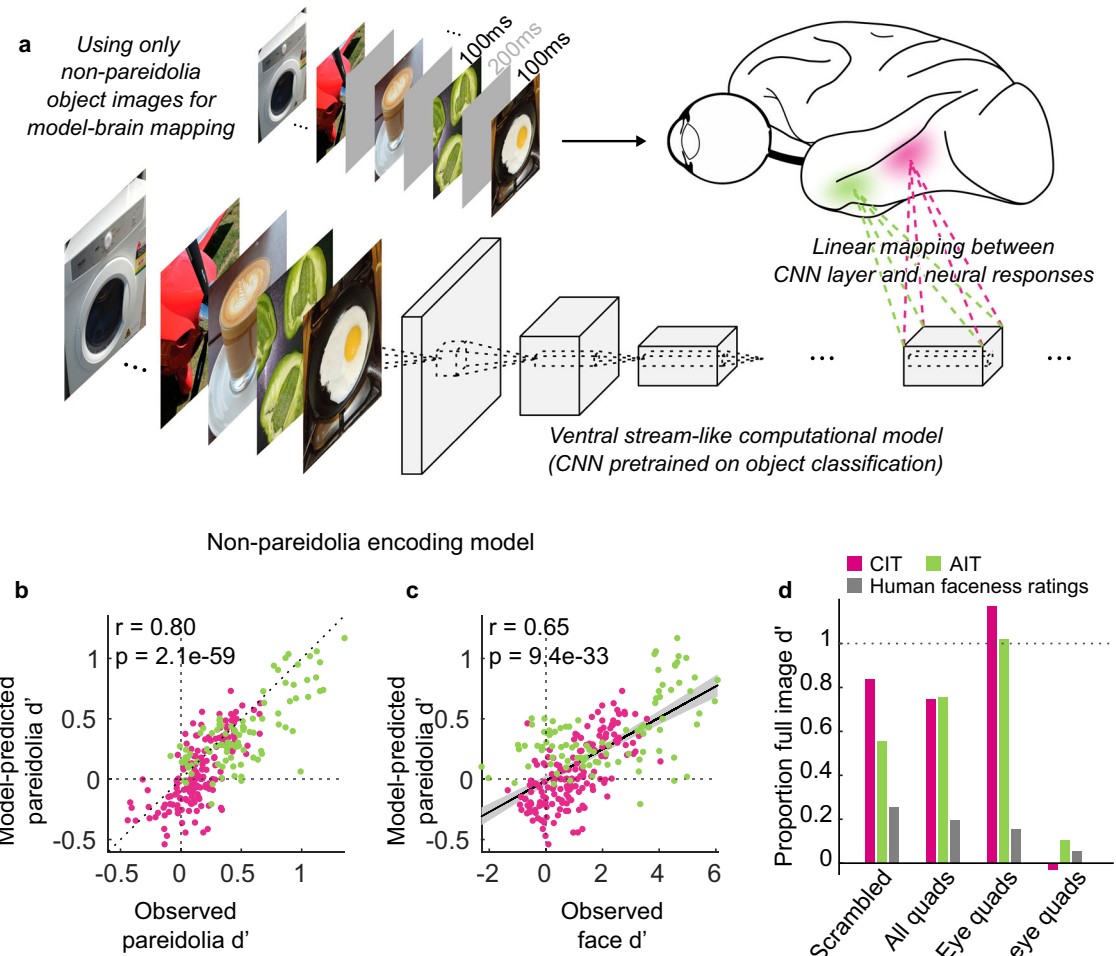

**Fig. 7 | Pareidolia selectivity is predicted by the feature-tuning estimated from non-pareidolia, non-face images. a** Design of the non-pareidolia encoding model. This encoding model was based on a CNN trained on object classification. We input into this model the same images that were presented to the monkeys. Using cross-validated subsets of only the 100 matched controls and 40 non-face objects, we estimated a linear mapping from the model to each neural site (see "Methods"). This resulted in an encoding model for each neural site that captures the tuning for characteristics of only non-pareidolia, non-face images. **b** Scatterplot showing the similarity between each neural site's observed (i.e., computed from neural responses) pareidolia $d'$ and its corresponding model-predicted value. Each dot represents a neural site in central IT (CIT; pink; $n = 171$) or anterior IT (AIT; green; $n = 95$). The values on the top left corner depict the Pearson's correlation and the corresponding $p$-value calculated using the corr function in MATLAB (R2021b). **c** Scatterplot showing the correlation between

neural face selectivity (observed face $d'$) and model-predicted pareidolia selectivity (pareidolia $d'$). The black line indicates an ordinary least squares (OLS) linear regression fit, with shaded 95% confidence intervals error bands. Same conventions as (**b**). **d** The proportion of mean $d'$ value (relative to the $d'$ of the full original images) for all quadrants, eye quadrants only and non-eye quadrants only, computed based on the responses of the non-pareidolia encoding models of face-selective units (observed face $d' > 1$) in CIT (pink bar; $n = 80$), AIT (green bar; $n = 58$), and human subjects' faceness ratings (gray bar; $n = 100$) for a subset of 34 images. Example images shown in (**a**). adapted from Wardle, S.G., Taubert, J., Teichmann, L. et al. Rapid and dynamic processing of face pareidolia in the human brain. Nat Commun 11, 4518 (2020). https://doi.org/10.1038/s41467-020-18325-8 released under a CC BY license: https://creativecommons.org/licenses/by/4.0/.

isolated eye-like features that merely represent object parts without the context of a face-like configuration.

Holistic processing of a facial configuration is thought to play a fundamental role in face perception[9–11,40]. In support of this notion, we show that shuffling the global configuration of pareidolic images lowers perceived faceness to the level of matched control objects. Human observers who indicated that they mentally reshuffled the parts back to their original configuration showed higher levels of perceived faceness for pareidolia images. These findings confirm that a face-like configuration underlies the illusory perception of faces in objects. In contrast, macaque face cells remained pareidolia selective (albeit significantly less so) even when the global configuration of the pareidolia images was scrambled. This selectivity persisted even for scrambled versions of images for which the monkeys had never seen

the original version (Supplementary Fig. 1), indicating that it was not driven by image familiarity. Moreover, the observed reduction in pareidolia selectivity was not restricted to face cells, which are thought to underlie face processing[3,7,18–20,23,43,59]. Instead, we observed a general reduction in all units in CIT and AIT, though reduction was smaller for the nonface units than for the face units in CIT. A potential explanation is that, compared to AIT, the pareidolia selectivity for the original images of nonface units in CIT was lower (CIT: 0.04, AIT: 0.3) and therefore there was less potential for reduction because of scrambling. Nevertheless, the fact that scrambling equally (AIT) or almost equally (CIT) affected all units, independent of their face selectivity, suggests that pareidolia selectivity in face cells is not explained solely by the face-like configuration of the object features. That is, the reduction in selectivity occurs due to a general breakdown of configuration, rather

than the face-like configuration specifically. Overall, our results suggest a significant contribution of local features in driving face cells.

The notion that local features drive neural pareidolia selectivity was reinforced by our results from presenting individual image parts. A shortcoming of using scrambling as a method to investigate holistic processing is that it does not distinguish between breaking up the configuration and changing the absolute position of object parts[60]. Moreover, scrambling can introduce a new, potentially confounding, configuration (see Supplementary Fig. 2). When presenting individual quadrants in isolation, we found that 75–80% of the variance and magnitude in pareidolia selectivity was explained by local features. This experiment not only confirmed but strengthened the results of the scrambling experiment. That is, face-cell responses did not require a holistic, face-like configuration to remain pareidolia selective, and instead encode local features. In this way, face cells are no different from other category-selective IT neurons, which have been shown to encode local features or object parts, rather than a global spatial arrangement of object features[60–64].

What could these local features that mostly drive pareidolia selectivity in face cells be? Our results show that the visual features underlying pareidolia selectivity are mostly contained in the eye quadrants of the illusory faces. The significance of eyes for face cells has previously been demonstrated[23,25,28,29]. Similarly, a reduced (but not abolished) selectivity has also been demonstrated for scrambled real faces[3,5] (as well as in our current manuscript, Supplementary Fig. 3), which retain individual eye parts. However, in those cases, visible individual face parts were still indicative of the presence of a face. Pareidolia images do not contain such a confound: the "eyes" of an illusory face are simply object parts when presented in isolation. Arguably, an egg yolk in isolation does not resemble an eye more so than it resembles an actual egg yolk. A more parsimonious explanation, therefore, is that face cells do not require actual face parts for selective responses, and that local object features, or object combinations, are enough to drive their selectivity[8,42,65,66]. Moreover, these local object features may not be intuitively interpretable[42], such as in the case of the faceness matched subset (Fig. 3b), where object parts that represent pareidolic eyes are not clearly discernible.

Overall, our results indicate that face-cell tuning is driven largely by relatively lower-level features such as contrast or shape, as opposed to a tuning driven by a face-like configuration. Such lower-level features are present in object parts as well as in actual eyes and possibly contribute to the detection of a face[35]. Some evidence for this has been observed in causal studies where the effect of stimulating face and body-selective IT neurons is not limited to the preferred categories, but also affects the perception of other categories, such as non-faces for face cells and faces and houses for body cells[20,67]. These ideas are supported by our own results showing that pareidolia selectivity was predicted by the CNN encoding model, which was pre-trained on general object classification and fit on neural responses to only non-faces. Taken together, our results provide support for the notion that, rather than encoding high-level, category-specific features, neurons in IT cortex owe their category selectivity to learnt combinations of lower-level, domain-general features that together form an integrated object space[42,68–72].

A relatively smaller 20–25% of pareidolia selectivity could not be explained by the eye quadrants in isolation, suggesting a smaller contribution of additional factors. One possibility is that there was a significant contribution of the face-like configuration in pareidolia images. Indeed, one result we observed that supports this possibility is that face selectivity was negatively correlated with object rank correlation (a metric we used to assess if neural responses were selective to features shared between pareidolia images and their matched control objects; Supplementary Fig. 4a, b). Given that pareidolia images and matched controls share features associated with object identity (such as shape, color, or texture), this result suggests that something other

than these shared features drives the face-cell responses, possibly the presence of the face-like configuration[36]. However, this could also be the arrangement of smaller subsets of parts, such as two pareidolic eye features, and not the whole face-like configuration per se[25]. Indeed, several of our results point to a contribution of characteristics that are not face-like. First, the negative correlation between face selectivity and object rank correlation was preserved even after scrambling the face-like configuration (Supplementary Fig. 4c, d). Second, configuration scrambling reduced pareidolia selectivity not just in face cells, but in all units, suggesting a contribution of features not correlated with faces. These features could be nonface-like configurations of object parts, as well as larger properties of the whole object upon which the face-like arrangement is embedded (e.g., the large round frying pan, fruit, etc.). Finally, the result that pareidolia selectivity was lowest for scrambled images (compared to summing across all the individual quadrants) points to a negative effect of scrambling, particularly in CIT (Supplementary Fig. 2). This suggests that face cells are at least modulated by nonface-like configurations. Thus, further research will be necessary to disentangle the potential contribution of face-like configurations in pareidolia selectivity from other nonface-like characteristics hinted at by our results.

Regardless of the configuration, objects can vary in perceived faceness. Any object, such as an apple or a house, could be evaluated on how face-like it is[72], without necessarily creating the perceptual illusion of a face. However, this general faceness of an object is distinct from the notion of face pareidolia: the fact that a round object is generally rated higher in faceness than a square object does not imply the presence of an illusory face in the round object. Indeed, our human faceness ratings highlight that the phenomenon of face pareidolia occurs specifically for visual patterns with a spurious face-like arrangement, which can be separated from the more general faceness of an object. The fact that this was different for neurons indicates a disconnect between what we as humans see in the pareidolia images (an illusory face) and what face cells respond to (object parts).

It remains an open question if our results reflect a discontinuity between perception and neural responses, or a discontinuity between species. Though humans consistently report perceiving illusory faces in pareidolia images, even assigning age, gender, and emotion to these nonface objects[33,34], it is unclear whether monkeys even perceive illusory faces. While monkeys cannot be asked whether they perceive an illusory face, a behavioral looking preference for face pareidolia has been previously demonstrated in monkeys[23,31]. However, it is unclear whether a viewing preference implies a perceived face illusion. That is, while monkeys tend to direct their gaze towards faces[23,31,73–75], they also tend to fixate small round object parts without the presence of an illusory or real face[74]. Thus, a behavioral preference for illusory faces could simply be a preference for small round shapes, which are more prevalent in illusory faces, without necessarily indicating that the monkeys perceive illusory faces. In fact, given this behavioral preference, it is even possible that monkeys, unlike humans, rely more on local features, and perceive the individual object parts equally as face-like as the full pareidolia image[16,76,77]. Thus, our finding that the pareidolia selectivity of face cells is driven primarily by object parts could reflect an interspecies difference. On the other hand, our results could reflect the neural properties only of the 8 monkeys we tested here, which is not a large sample. However, our key results are replicable and significant not just across neurons, but also across monkeys (Supplementary Tables 1 and 2, Supplementary Fig. 8), allowing us to draw population-level inferences[78].

It is also possible that our results are constrained by how the stimuli we used to examine monkey face cell responses were selected and rated by humans. However, this reflects a broader methodological challenge inherent in cross-species research. There is a long tradition in visual neuroscience to rely on human perceptions to design and interpret neural responses. While the reliance on human-selected

stimuli is a necessary tool, our results highlight that neural activity may not always reflect what we as humans see in these images. Indeed, these considerations become particularly relevant when dealing with ambiguous stimuli that represent complex phenomena like face pareidolia or when dealing with a stimulus category such as faces, for which researchers have a strong perceptual bias. Further, while humans do rely on a holistic configuration for perceptual pareidolia, future experiments could test if face-selective regions in humans are sensitive to the non-holistic attributes of the pareidolia images.

Finally, our inclusion of the face-deprived monkeys in the current study might seem counterintuitive to the hypothesis we tested. Previous work from our lab has shown that monkeys reared without experiencing faces for the first year of their life do not develop face-selective domains in the IT cortex[73]. However, we decided to include them since both monkeys had more than 2 years of experience with faces when we ran these experiments and fell within the range of face $d'$ demonstrated by the control monkeys (Supplementary Fig. 5, $t7 = 0.08$, $p = 0.94$; Supplementary Fig. 6). Importantly, excluding them from our analysis did not qualitatively change any of our results.

In conclusion, we show that face cells in IT cortex are pareidolia selective and that pareidolia selectivity is correlated with face selectivity. We found that human faceness ratings did not explain the response variance of macaque face units. Moreover, though scrambling the face-like configuration of the face pareidolia images did impact behavioral faceness ratings by human subjects, it did not abolish the pareidolia selectivity of face cells. Instead, macaque neuronal pareidolia selectivity was primarily driven by the individual object parts that represent "eyes" in the illusory face, unlike human perceptual pareidolia, which is primarily driven by the holistic configuration. Finally, the pareidolia selectivity of face cells could be captured by an encoding model fit on non-face, non-pareidolia images. Taken together, our results indicate that face cells encode local visual features, without requiring the spatial arrangement of the features, contrary to human visual perception, which requires holistic information[9,60,61,63,79]. This suggests that macaque face cells encode object features similarly to other category selective neurons in IT[27,80] and provides further evidence for the notion that face-cell selectivity is not restricted to faces but is instead indicative of a broader tuning in an integrated object space[8,36,42,69,71]. More generally, our results underscore the importance of exercising caution when interpreting neural activity, regardless of the species, as it may not always mirror our conscious perceptual experiences. It is tempting to understand selective responses in terms of what we, as human observers, see. However, in doing so, one might end up imposing the biases of human perception itself, such as seeing a face where none exists, onto neural activity.

## Methods

All procedures were approved by the Harvard Medical School Institutional Animal Care and Use Committee and conformed to NIH guidelines provided in the Guide for the Care and Use of Laboratory Animals.

### Subjects and array location

Seven male Macaca mulatta (5–13 kg; 2–17 years old) and one male Macaca nemestrina (15 kg, 15 years old) implanted with floating microelectrode arrays (32 channels, MicroProbes, Gaithersburg, MD or 128 channels, NeuroNexus, Ann Arbor, MI) or microwire bundles (64 channels; MicroProbes) were used in this experiment. The arrays were chronically implanted based on fMRI localization or anatomical "bumps"[81] in the lower bank of the superior temporal sulcus, at the location of the middle face region (ML) for four monkeys and at the location of anterior face region (AL) for five monkeys (one monkey had arrays in both middle and anterior face region). Two of the monkeys (M2 with an array in central IT, and M4 with arrays in both central and anterior IT) were face-deprived, i.e., they were prevented from seeing

faces for the first year of their life (similar procedure, but different monkeys than the ones used in ref. 73). However, at the time that these experiments were run, both monkeys had been exposed to faces for over 2 years.

### Visual experiments

The monkeys sat upright in a plastic monkey chair, facing an LCD display screen 51 cm in front of the monkey. They were given continuous juice reward for maintaining fixation on a red dot ($0.2 \times 0.2°$) in the center of the screen. Eye movements were tracked at 120 Hz using ISCAN system (Woburn, MA, http://www.iscaninc.com/). After array implantations, the receptive field of each array was mapped by presenting faces and fractals. The images were presented using the software MonkeyLogic (https://monkeylogic.nimh.nih.gov/) at the rate of 100 ms on, 200 ms off, while the monkeys fixated the red dot in the center of the screen. For the main experiments, we used a rsvp-style task paradigm, where images (5° by 5°) were presented on a gray background at the rate of 100 ms on, 200 ms off at the center of the array's mapped receptive field.

### Stimuli

The 100 pareidolia images used in Experiment 1, each with their matched controls, and 100 real faces on a natural background (68 monkey and 32 human faces), were obtained from Dr Jessica Taubert (some used in refs. 31,33,34). The 32 human faces are the same as used in ref. 33 and are available at https://osf.io/9g4rz. We also presented quadrant scrambled versions of the pareidolia, matched control and real face images. To do this, we divided the image into four equal quadrants and randomly shuffled the location of the quadrants (Fig. 4b, Supplementary Fig. 2a). In Experiment 2, to investigate the effect of familiarity (Supplementary Fig. 1), we presented only the quadrant scrambled versions of 119 pareidolia images and their matched controls (taken from ref. 34) that the monkeys had never seen before. In Experiment 3, to compare the neural responses with the human faceness ratings, we presented the monkeys with all 200 pareidolia images and their matched controls from ref. 34 (the images they used in their experiment 2A; images and data available at https://osf.io/f74xh/). Finally, in Experiment 4, we took the top 50 images based on the neural response from Experiment 1 and presented the original and scrambled versions of the full images. To ensure the shared presence of edges in both original and scrambled images, we introduced a gray cross in both sets of images. To ensure that the gray cross did not obscure any part of the full image, the quadrants were moved slightly away from each other in the full images. We also presented the four quadrants in isolation for both original and scrambled images (see Fig. 6a, Supplementary Fig. 2a for examples). The gray background in the quadrant-only images was the same color as the screen background. For the analysis presented in Fig. 6, we selected a subset of 34/50 images where the "eyes", and "non-eyes" were clearly separable. We used the same 34 images for the human faceness ratings task, though in their original format, without the gray cross. Each neural recording experiment additionally included 40 faces (20 human faces, 20 monkey faces) and 40 non-faces (20 familiar, 20 nonfamiliar) on a white background (Fig. 1a), used to quantify the selectivity of the neural units. The 20 non-familiar images of nonfaces are from ref. 82, presented in ref. 83 as well, while the 20 images of nonfaces familiar to the monkeys, human and monkey face images were taken in our lab.

### Human behavioral experiment

A total of 100 human subjects (mean age = 31.8 y, SD = 10.7) were recruited from an online platform Prolific. The participants were then redirected to our own website where we used jsPsych, a JavaScript-based framework, to run the experiment. All participants provided informed consent and received monetary compensation for participation in the experiment. The experiment was conducted according to

protocols approved by the Institutional Review Board at Harvard Medical School. We used a base set of 34 images where the "eyes", and "non-eyes" were clearly separable (from monkey Experiment 4; see above). From this subset of 34, we selected images from 6 categories—original pareidolia, original matched control, scrambled pareidolia, scrambled matched control, pareidolia quadrants and matched control quadrants. From each category, we randomly selected 10 pareidolia images and their 10 matched controls, 10 scrambled pareidolia images and their 10 scrambled matched controls, 12 pareidolia quadrants and their 12 matched control quadrants (3 from each of the four positions, i.e., 3 upper left quadrants, 3 upper right, 3 lower left and 3 lower right). We additionally included 44 additional pareidolia images randomly selected from the 200-image set from ref. 34 and 2 real faces. Thus, we presented a total of 110 images per subject. All images were scaled to be the same size and presented in the center of the screen, including the quadrants. Therefore, we did not need to present the quadrants in their original and scrambled locations as for the monkey experiments. The random image assignments per subject were implemented using a custom Matlab R2021b script, so no participants saw the scrambled versions or individual quadrants of an original pareidolia image they had already seen. We kept the instructions and task as similar to[34] as possible. Briefly, the experiment began with an instruction screen that included example images of the type of stimuli the subjects would see. Subjects were instructed to report how easily they could see a face in the image as it was presented to them. During the experiment, each image remained present until the participants used a slider to indicate a response. Under each image, we presented the question "Rate how easily you can see a face in this image" along with a slider ranging from 0 ("cannot see a face") to 10 ("easily see a face"). At the end of the experiment, the subjects were asked a post-task question "Some of the images that you saw appeared jumbled. How frequently did you try to mentally shift the parts to make them whole again?", with the following answer choices as a response: "Never, rarely, sometimes, most of the time, always". Out of 100 subjects, 11 responded "never", 15 responded "rarely", 29 responded "sometimes", 30 responded "most of the time", and 15 responded "always".

## Non-pareidolia encoding model

The methods used for the CNN encoding model were largely the same as those in ref. 42. Briefly, we used an ImageNet-trained AlexNet model[58] as a base. We then fit a linear mapping between the activations of a chosen model layer and the trial-averaged neural responses for each neural site. We used the same model layer for all neural sites, by selecting the one that on average most effectively predicted responses to non-pareidolia images. For this linear mapping, we used only matched controls ($n = 100$) or non-face object ($n = 40$) images, thus excluding the pareidolia image ($n = 100$). This resulted in a non-pareidolia encoding model that quantifies neural tuning exclusively based on features present in non-pareidolia images.

We fit the linear mapping between model activations and neural responses: First, we normalized the CNN layer's outputs per channel using the SD and mean across all pareidolia, matched control, and non-face object images (and across locations for convolutional layers). We then reduced the dimensionality through principal component analysis. Next, we fit a linear support vector regression model, using 35-fold cross-validation, to map neural responses onto the principal components of the normalized CNN activations (using the MATLAB 2020a function 'fitrlinear', with the SpaRSA solver and regularization parameter lambda set to 0.01). Before fitting, we centered the predictors on the mean of the training fold, and both centered and standardized neural responses using the mean and SD of the training fold. We then evaluated performance based on all concatenated out-of-fold predicted responses. For pareidolia images, which were excluded from the training folds, we calculated the image-wise average across out-of-

fold predictions. Finally, we determined predicted pareidolia selectivity by calculating the pareidolia $d'$ using these out-of-fold predicted responses. For the data shown in Supplementary Fig. 7, we input both pareidolia and control images into the non-face encoding model of ref. 42, which was fit on an entirely independent dataset. Briefly, this independent model was constructed using a CNN-based encoding model (using the same base model as above) and was fit (linearly mapped) on the responses from 449 central IT sites to 932 exclusively inanimate, non-face objects (for more details see ref. 42). None of the pareidolia images or matched controls in the present study were shown in this prior experiment with separate data and protocols.

## Data analysis

**Firing rates.** Neural signals were amplified and sampled at 40 kHz using a data acquisition system (OmniPlex, Plexon, Dallas, TX). Multi-unit spiking activity was detected using a threshold-crossing criterion. The neural response was defined as the spike rate in a 150 ms time window starting at 30–100 ms after the image onset, depending on the monkey, i.e., M3 (CIT): 30–180 ms; M8 (AIT): 100–250 ms; all remaining monkeys: 50–200 ms. The response onset was decided by manually inspecting the time course of the neural response averaged across all units per monkey. The firing rates per neural site were trial averaged per image.

**Response reliability.** Firing-rate reliability per neural site was used as a selection criterion for inclusion of neural units in further analysis. To calculate the reliability of the units, the number of repeated presentations or trials for each image was randomly split in half. Then two response vectors, one per half of the trials, were generated by trial averaging the neural response, and the correlation between these two split-half response vectors was calculated. This step was repeated 100 times, each time using different random splits, and an average correlation $r$ was computed. The reliability $\rho$ was computed by applying the Spearman-Brown correction the average correlation $r$:

$$\rho = \frac{2r}{(1+r)} \qquad (1)$$

Only neural units with a split-half reliability $\rho > 0.4$ were included in further analysis. For experiment 1, this resulted in the inclusion of 64/64 units from M1, 73/128 units from M2, 31/64 units from M3 and 40/64 units from M4, leading to a total of 208 recorded units in central IT. For anterior IT, 60/64 units from M5, 13/64 units from M6, 24/64 units from M7, 57/64 units from M8 and 9/64 from M4, totaling 163 neural units passed this criterion. For experiment 2, 58/64 units from M1, 50/128 units from M2, 20/64 units from M3 and 34/64 units from M4 passed criterion, leading to a total of 162 recorded units in central IT and 49/64 from M5, 18/64 units from M7, 45/64 units from M8 and 2/64 from M4, totaling 114 neural units in anterior IT. For experiment 3, this criterion included 58/64 units from M1, 56/128 units from M2, 23/64 units from M3 and 34/64 units from M4, leading to a total of 171 recorded units in central IT and 64/64 from M5, 4/64 units from M6, 19/64 units from M7, 54/64 units from M8 and 3/64 units from M4, totaling 144 neural units in anterior IT. Finally, for experiment 4, we included 41/64 units from M1, 52/128 units from M2, 27/64 units from M3 and 25/64 units from M4, leading to a total of 145 recorded units in central IT and 10/64 from M5, 12/64 units from M6, 1/64 units from M7, and 49/64 units from M8, totaling 72 neural units in anterior IT.

For computing the 95% bootstrapped CI and mean across neural sites of the noise ceiling (depicted by gray boxes and white line in Fig. 3d), the neural response reliability was multiplied by the reliability of the faceness-rating. In this case, the neural response reliability was computed as the rectified Pearson correlation between odd and even trial-averaged responses, with a Spearman-Brown correction (formula for Spearman-Brown correction shown in Eq. 1). For the faceness

ratings, reliability was computed as the Pearson correlation between odd and even subject-averaged faceness ratings, with a Spearman-Brown correction.

**Face selectivity.** We quantified face selectivity using a d′ index, which compared trial-averaged responses to faces and to non-faces:

$$d' = \frac{\mu_F - \mu_{NF}}{\sqrt{\left(\sigma_F^2 + \sigma_{NF}^2\right)\frac{1}{2}}} \tag{2}$$

where $\mu_F$ and $\mu_{NF}$ are the across-stimulus averages of the trial-averaged responses to faces and non-faces, and $\sigma_F$ and $\sigma_{NF}$ are the across-stimulus standard deviations. We used the same d′ metric to quantify pareidolia selectivity, which was computed based on neural responses to pareidolia images and matched controls.

**Statistical inference.** P-values were calculated using paired t-tests when comparing differences between conditions within face units, and unpaired t-tests when comparing face and nonface units. For $R^2$ and correlations permutation testing was performed by randomly shuffling one of the two variables. For the paired difference between two correlations, the condition labels were randomly shuffled for each pair of observations. 95% confidence intervals were calculated using the bias corrected accelerated bootstrap[84], based on 10,000 iterations.

### Reporting summary
Further information on research design is available in the Nature Portfolio Reporting Summary linked to this article.

## Data availability
All correspondence and material requests should be addressed to the corresponding author (S.S). Data generated and used in this study have been deposited on the Open Science Framework (OSF) database (https://osf.io/z2kra/). The data are available under full access, access can be obtained by downloading the.mat files. All data necessary to reproduce the figures in this paper are publicly available in the OSF database (https://osf.io/z2kra/). Source data are provided with this paper.

## Code availability
The data were analyzed using custom MATLAB code (R2021b). The custom-written code necessary to reproduce the data figures in this paper are publicly available in the OSF database (https://osf.io/z2kra/).

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

## Acknowledgements

We thank Dr. Jessica Taubert and Dr. Susan Wardle for sharing stimuli and for suggestions relating to the human behavioral task. We also thank Dr. Will Xiao for helpful feedback and comments on the manuscript. This work was supported by the Harvard Lefler Fellowship, Hearst Fellowship and Gordon Fellowship (to S.S.), Alice and Joseph Brooks Fund Postdoctoral Fellow (K.V.), NIH grant EY16187 and P30 EY012196 and R01 EY025670 (to M.S.L.).

## Author contributions

Conceptualization: S.S., K.V., and M.S.L. Investigation: S.S and A.V.J. Formal analysis: S.S. and K.V. Methodology: S.S., K.V., and A.V.J. Visualization: S.S and K.V. Supervision: K.V. and M.S.L. Writing—original draft: S.S. Writing—review and editing: S.S., K.V., A.V.J., and M.S.L.

## Competing interests

The authors declare no competing interests.
