## [Transparent Peer Review file · Nature Communications]

Face cells encode object parts more than facial configuration of illusory faces.

Corresponding Author: Dr Saloni Sharma

This manuscript has been previously reviewed at another journal. This document only contains reviewer comments, rebuttal and decision letters for versions considered at Nature Communications.

Version 0:

Reviewer comments:

Reviewer #1

(Remarks to the Author)

The authors have substantially revised the manuscript and addressed most of my concerns. I have one remaining question:

Figure 7 suggests that the "non-pareidolia encoding model" and the "non-face encoding model" are able to predict the pareidolia selectivity of IT neurons. Since the authors suggest that monkey IT neurons rely heavily on local features to process illusory faces, which is inconsistent with human perception, I couldn't help but wonder whether the models behave similarly to monkey neurons or to human perception. This can be addressed by examining the responses of the models to scrambled images and isolated quadrants, as the authors did earlier in the manuscript for IT neurons and human behaviour. Such an analysis will also deepen our understanding of the computational mechanisms underlying face pareidolia.

Reviewer #2

(Remarks to the Author)

The authors have strengthened the manuscript by including additional human behavioral data and clarifying aspects of the manuscript. It is a well conducted study and the data are interesting, but I remain confused as to what the authors are really trying to say. I'm not sure what I should take away from the manuscript or why it's important.

When I reviewed the original manuscript I thought the authors were trying to say that how face-selective cells respond to pareidolic faces is different from how they respond to real faces. This is why I thought it was important to show data for scrambled faces. But based on the authors' responses it doesn't seem this is the point they were trying to make. If the authors are just characterizing how face-selective cells respond to pareidolic faces, then the manuscript is OK, but I don't know what insight the data really provide. Why is it important to understand how face-selective cells respond to pareidolic faces? If the pattern of response to the scrambling manipulations is similar for pareidolic and real faces, which it seems to be, then what do the pareidolic results tell us beyond the data for real faces? The manuscript would be stronger if the authors can provide more clarity.

More generally, I remain concerned that much of the manuscript relies on comparing neural data from monkeys with human behavioral data. This is particularly important for this study because the perception of a face in face pareidolia is highly subjective – there is no real face in the stimulus. How well should we expect human subjective ratings to compare with monkey perception? Further, it is important to remember that there is substantial variability in human ratings of face pareidolia – a species difference could account for some of the results here, but so could individual variability, even if the species are well aligned, given the relatively small number of monkeys tested here. I think the authors need to be very cautious about drawing strong conclusions from these comparisons across monkeys and humans.

The authors ultimately argue that their results show that face pareidolia selectivity "was not driven by the face-like configuration" but by local features. But it strikes me that their results show that both configuration and local features contribute - after all quadrant scrambling reduces face pareidolia selectivity and the eye quadrants do not account for all pareidolia selectivity. The authors can argue that the relative contribution of local features is greater than the contribution of

configuration, but I don't think it's either/or.

Minor

- How many participants were in the different 'mental unscrambling groups'?
- What is shown in Figure 3c? I thought this was a comparison between faceness matched pareidolia and control images but the x-axis is labelled "face d' <1 and face d' >1". This doesn't seem to match the reference to this panel in the text.

Reviewer #3

(Remarks to the Author)

I thank the authors for their detailed response. I believe genuine effort has been made to address all concerns. I have no further comment.

In this document, the original reviewer comments are in blue, our responses in red, lines from the manuscript in black.

Reviewer #1 (Remarks to the Author):

The authors have substantially revised the manuscript and addressed most of my concerns. I have one remaining question:

Figure 7 suggests that the "non-pareidolia encoding model" and the "non-face encoding model" are able to predict the pareidolia selectivity of IT neurons. Since the authors suggest that monkey IT neurons rely heavily on local features to process illusory faces, which is inconsistent with human perception, I couldn't help but wonder whether the models behave similarly to monkey neurons or to human perception. This can be addressed by examining the responses of the models to scrambled images and isolated quadrants, as the authors did earlier in the manuscript for IT neurons and human behaviour. Such an analysis will also deepen our understanding of the computational mechanisms underlying face pareidolia.

We thank the reviewer for this interesting suggestion! We have now updated our Figure 7 to include a panel d, which compares model responses to the scrambled and isolated quadrants to human faceness ratings and shows that the responses of the non-pareidolia model units are primarily driven by the eye quadrants in the illusory faces, like macaque face cells and inconsistent with human perception.

Reviewer #2 (Remarks to the Author):

The authors have strengthened the manuscript by including additional human behavioral data and clarifying aspects of the manuscript. It is a well conducted study and the data are interesting, but I remain confused as to what the authors are really trying to say. I'm not sure what I should take away from the manuscript or why it's important.

When I reviewed the original manuscript I thought the authors were trying to say that how face-selective cells respond to pareidolic faces is different from how they respond to real faces. This is why I thought it was important to show data for scrambled faces. But based on the authors' responses it doesn't seem this is the point they were trying to make. If the authors are just characterizing how face-selective cells respond to pareidolic faces, then the manuscript is OK, but I don't know what insight the data really provide. Why is it important to understand how face-selective cells respond to pareidolic faces? If the pattern of response to the scrambling manipulations is similar for pareidolic and real faces, which it seems to be, then what do the pareidolic results tell us beyond the data for real faces? The manuscript would be stronger if the authors can provide more clarity.

We thank the reviewer for their comments and agree that it is important to clarify the rationale behind studying face cell responses to pareidolia faces. In terms of what the scrambling manipulation tells us, we would like to point out (as we mentioned in the previous round of comments to the reviewer) that the results of our Experiment 4 (where we presented isolated quadrants) tell us that the scrambling manipulation may not be the best technique to glean configural processing in IT face cells. This is because the neural response is negatively affected by the spurious configuration introduced by this approach, particularly in CIT (see our Supplementary Fig. 2, comparing sum of quadrants to full quadrant scrambled images).

We would also like to clarify that the main goal of our study is not merely to establish how face-selective cells respond to pareidolic faces, but to use pareidolic faces as a tool to understand the general neural mechanisms of face detection. Moreover, the key reason we believe it is important to understand how face cells respond to pareidolia faces is because as the reviewer themselves points out - *there is no real face in the stimulus*. Thus, using such images allows us to better investigate holistic processing in face cells without the confound of the presence of recognizable face parts (scrambled or isolated). We already allude to this in the introduction, "The importance of the face-like configuration is stressed by the fact that normal facial features are absent in these images...". To further clarify our rationale, we have made the following additions/changes to the introduction-

"These responses, **thought to underlie rapid face detection**, are suggested to be driven primarily by a template-matching process, where the template represents a rudimentary form of low or mid-level visual features that approximate a face-like configuration^{8, 33, 35, 39-41}"

"Are responses to illusory face stimuli informative about the processing of actual faces? Recent evidence suggests that non-face stimuli engage the same mechanisms in macaque face cells as actual faces do⁴². The fact that macaque face selective regions respond to illusory faces indicates that these stimuli can be used to study general face processing mechanisms at the neural level, without assuming that macaques experience the illusion as humans do. Moreover, since these stimuli activate face-selective systems without the confounding presence of recognizable face parts, they offer unique insights into holistic processing in face cells."

More generally, I remain concerned that much of the manuscript relies on comparing neural data from monkeys with human behavioral data. This is particularly important for this study because the perception of a face in face pareidolia is highly subjective – there is no real face in the stimulus. How well should we expect human subjective ratings to compare with monkey perception? Further, it is important to remember that there is substantial variability in human ratings of face pareidolia – a species difference could account for some of the results here, but so could individual

variability, even if the species are well aligned, given the relatively small number of monkeys tested here. I think the authors need to be very cautious about drawing strong conclusions from these comparisons across monkeys and humans.

We appreciate the reviewers' concerns about drawing strong conclusions based on monkey-human comparisons and the challenge of comparing human subjective ratings with monkey perception. Note that we already refer to this point as a limitation in the Discussion ("It is also possible that our results are constrained by how the stimuli we used to examine monkey face cell responses were selected and rated by humans..."). Regarding the reviewers point of substantial variability in human ratings of face pareidolia, while there may be some subjectivity in which exact images evoke the perceptual experience of a face, the phenomenon of face pareidolia itself (that is, the experience of seeing faces in objects) has been consistently and robustly empirically demonstrated by ours and other groups for the set of images we used in our experiments (Wardle et al, 2022; Taubert et al, 2017). This makes this image set suitable for studying how macaque face cells respond to illusory faces, regardless of individual variability in human ratings.

Finally, regarding sample size, we acknowledge that 8 is not a large sample, but small sample sizes are a universal limitation in monkey studies. Moreover, our sample size of 8 monkeys (or even 4 monkeys per region) exceeds the typical standard of $n = 2$ in the field (In awake monkey neurophysiology, there is the saying: "One monkey is an anecdote, two monkeys are perpetual truth." Fries & Maris, J cog Neurosci, 2022, What to do if N is Two?: Azadi et al., 2024; Chang & Tsao, 2017; Duyck et al., 2021; Esmailpour & Vogels, 2024; Fries & Maris, 2022; Inagaki et al., 2023; Kar & DiCarlo, 2021; Rose & Ponce, 2024; Tsao & Livingstone, 2008; Waidmann et al., 2022; Yang & Freiwald, 2023; Yao & Vanduffel, 2024; Zhang et al., 2024). Importantly, our key results are replicable and significant not only across neurons, but also across monkeys (as shown in our Supplementary Table 1 and newly added Supplementary Table 2 and Supplementary Fig. 8), which does allow us to make population-level inferences (Fries and Maris, 2022). To address this comment, we have added a new Supplementary Table 2, which shows the positive correlation between pareidolia selectivity for eye-parts and full pareidolia images per monkey and a Supplementary Fig. 8, which shows the contribution of isolated quadrans in driving the face-cell pareidolia selectivity per monkey.

Supplementary Fig. 8. The contribution of isolated quadrans in driving the face-cell pareidolia selectivity per monkey. a. The proportion of mean d' value (relative to the d' of the full original images) for all quadrants, eye quadrants only and non-eye quadrants only for central IT (CIT) of 4 monkeys is shown

in box plots. The black central horizontal line shows the mean across monkeys, the black central vertical line indicates confidence intervals, and the bottom and top edges of the box depict the 25th and 75th percentiles, respectively. Whiskers (vertical lines in color) show the most extreme data points not considered outliers. Grey circles connected by grey lines behind the box plots show the proportion mean d' per monkey. b. The proportion of mean d' value (relative to the d' of the full original images) for all quadrants, eye quadrants only and non-eye quadrants only for anterior IT (AIT) of 3 monkeys. Same conventions as in a. Note that in this figure, the proportion of the mean d' was calculated individually for each monkey based on that monkey's average pareidolia d' for all quadrants, eye quadrants only, non-eye quadrants only and the full pareidolia images.

Monkey ID (no. of units)	CIT		AIT	
	r	p	r	p
M1 (41)	0.54	0.0003		
M2 (FD; 52)	0.034	0.81		
M3 (27)	0.46	0.017		
M4 (FD; 25)	0.81	1.23×10^{-6}		
M5 (10)			0.49	0.15
M6 (12)			0.76	0.0042
M7 (1)			n/a	n/a
M8 (57)			0.72	4.9×10^{-9}
Mean	0.46		0.66	

Supplementary Table 2. Correlation between pareidolia selectivity for eye parts only and full pareidolia images. Table showing Pearson's correlation r between pareidolia d' for pareidolia eye parts only and full pareidolia images from Experiment 4 for each monkey in central (CIT; $n = 4$) and anterior IT (AIT; $n = 3$). One-sample t-tests on the Fischer-transformed correlation values indicate that this correlation was significant across monkeys (CIT: $t_3 = 2.52$, $p = 0.043$; AIT: $t_2 = 5.8$, $p = 0.014$). FD = face deprived.

Moreover, we have added the following in the Discussion -

“On the other hand, it is also possible that our results reflect the neural properties only of the 8 monkeys we tested here, which is not a large sample. However, our key results are replicable and significant not just across neurons, but also across monkeys (Supplementary Table 1 and 2; Supplementary Fig. 8), allowing us to draw population-level inferences⁷⁸.”

The authors ultimately argue that their results show that face pareidolia selectivity “was not driven by the face-like configuration” but by local features. But it strikes me that their results show that both configuration and local features contribute - after all quadrant scrambling reduces face pareidolia selectivity and the eye quadrants do not account for all pareidolia selectivity. The authors can argue that the relative contribution of local features is greater than the contribution of configuration, but I don't think it's either/or.

We thank the reviewer for giving us this opportunity to clarify our claim. Our claim for this result throughout the text has been “pareidolia selectivity of face cells is primarily driven by” which does imply a greater contribution of local features than configuration. Further, we acknowledge in our Discussion that the remaining 20-25% not explained by local features could be explained by the configuration—

“A relatively smaller 20-25% of pareidolia selectivity could not be explained by the eye quadrants in isolation, suggesting a smaller contribution of additional factors. One possibility is that there was a significant contribution of the face-like configuration in pareidolia images...”

Nevertheless, to address this comment, we changed the specific line that the reviewer pointed to in the abstract. Instead of face pareidolia selectivity “was not driven by the face-like configuration”, the line now says face pareidolia selectivity “did not require the face-like configuration”.

Minor

- How many participants were in the different ‘mental unscrambling groups’?

We have now added this information in the Figure legend of Fig. 4 where we show this data as well as in the methods section, –

“Out of 100 subjects, 11 responded ‘never’, 15 responded ‘rarely’, 29 responded ‘sometimes’, 30 responded ‘most of the time’, and 15 responded ‘always’.”

- What is shown in Figure 3c? I thought this was a comparison between faceness matched pareidolia and control images but the x-axis is labelled “face $d' < 1$ and face $d' > 1$ ”. This doesn't seem to match the reference to this panel in the text.

We have updated the figure legend of 3c to be clearer about what we plot there:

“Fig. 3. Faceness does not explain pareidolia selectivity in face cells. (a) Faceness-matched subset of the 10 least face-like pareidolia images (top row) and the 10 most face-like pareidolia images (bottom row) as rated by human subjects (taken from Wardle et al, 2022), corresponding to the darkened orange and blue dots in b. (b) Distribution of perceptual faceness ratings for all 200 pareidolia images (orange dots) and their matched controls (blue dots) taken from Wardle et al, 2022. Darkened dots indicate a faceness-matched subset of the 10 most face-like control images and the 10 least face-like pareidolia images depicted in a. Downward-pointing triangles indicate the means of the faceness-matched subsets. (c) Pareidolia d' values computed using the 20 faceness-matched image subset for nonface (face $d' < 1$, CIT = 84, AIT = 24) and face units (face $d' > 1$, CIT = 87, AIT = 120). Each marker represents a single neural site. Colored boxes: 95% bootstrapped CI, calculated by resampling sites; black line: mean. (d) Response variance for all 200 pareidolia images explained by faceness ratings. Black line: mean. Colored boxes indicating 95% bootstrapped CI are obscured by the black mean line. Grey boxes and the white line indicate the 95% bootstrapped CI and mean across neural sites of the noise ceiling (i.e., the maximum explainable variance, computed as the neural response reliability multiplied by the reliability of the faceness-rating). CIT = central IT; AIT = anterior IT.”

Reviewer #3 (Remarks to the Author):

I thank the authors for their detailed response. I believe genuine effort has been made to address all concerns. I have no further comment.

We thank the reviewer for their constructive feedback, which helped make our manuscript substantially stronger!